# Effects of Ethanol on Expression of Coding and Noncoding RNAs in Murine Neuroblastoma Neuro2a Cells

**DOI:** 10.3390/ijms23137294

**Published:** 2022-06-30

**Authors:** Mi Ran Choi, Sinyoung Cho, Dai-Jin Kim, Jung-Seok Choi, Yeung-Bae Jin, Miran Kim, Hye Jin Chang, Seong Ho Jeon, Young Duk Yang, Sang-Rae Lee

**Affiliations:** 1Laboratory Animal Research Center, Ajou University School of Medicine, Suwon 16499, Korea; mrchoi2007@ajou.ac.kr; 2Department of Pharmacy, College of Pharmacy and Institute of Pharmaceutical Sciences, CHA University, Pocheon 11160, Korea; jsy7122@naver.com (S.C.); cmb_jsh@naver.com (S.H.J.); 3Department of Psychiatry, Seoul St. Mary’s Hospital, College of Medicine, The Catholic University of Korea, Seoul 06591, Korea; kdj922@catholic.ac.kr; 4Department of Psychiatry, Samsung Medical Center, Seoul 06351, Korea; choijs73@gmail.com; 5Department of Laboratory Animal Medicine, College of Veterinary Medicine, Gyeongsang National University, Jinju 52828, Korea; ybjin@gnu.ac.kr; 6Department of Obstetrics and Gynecology, Ajou University School of Medicine, Suwon 16499, Korea; kmr5300@ajou.ac.kr (M.K.); zzanga-94@hanmail.net (H.J.C.); 7Department of Pharmacology, Ajou University School of Medicine, Suwon 16499, Korea

**Keywords:** ethanol, transcriptome profiling, lncRNA, *Cebpd*, *Rnu3a*

## Abstract

Excessive use of alcohol can induce neurobiological and neuropathological alterations in the brain, including the hippocampus and forebrain, through changes in neurotransmitter systems, hormonal systems, and neuroimmune processes. We aimed to investigate the effects of ethanol on the expression of coding and noncoding RNAs in a brain-derived cell line exposed to ethanol. After exposing Neuro2a cells, a neuroblastoma cell line, to ethanol for 24 and 72 h, we observed cell proliferation and analyzed up- and downregulated mRNAs and long noncoding RNAs (lncRNAs) using total RNA-Seq technology. We validated the differential expression of some mRNAs and lncRNAs by RT-qPCR and analyzed the expression of *Cebpd* and *Rnu3a* through knock-down of *Cebpd*. Cell proliferation was significantly reduced in cells exposed to 100 mM ethanol for 72 h, with 1773 transcripts up- or downregulated by greater than three-fold in ethanol-treated cells compared to controls. Of these, 514 were identified as lncRNAs. Differentially expressed mRNAs and lncRNAs were mainly observed in cells exposed to ethanol for 72 h, in which *Atm* and *Cnr1* decreased, but *Trib3*, *Cebpd*, and *Spdef* increased. On the other hand, lncRNAs *Kcnq1ot1*, *Tug1*, and *Xist* were changed by ethanol, and *Rnu3a* in particular was greatly increased by chronic ethanol treatment through inhibition of *Cebpd*. Our results increase the understanding of cellular and molecular mechanisms related to coding and noncoding RNAs in an in vitro model of acute and chronic exposure to ethanol.

## 1. Introduction

Alcohol is a toxic agent and an addictive drug, and chronic abuse causes structural and functional abnormalities in many organs including the brain and liver [1,2]. In particular, excessive alcohol use induces neurobiological and neuropathological alterations in several regions of the brain, such as the hippocampus and forebrain, through alterations of neurotransmitter systems, hormone systems, endogenous peptides, and neuroimmune processes, resulting in cognitive and learning deficits and neurodegenerative disease [3,4,5]. These numerous biological changes have increased interest in large-scale analyses designed to reveal the diverse mechanisms and pathways most prominent in alcohol abuse disorder.

With the development of high-throughput screening technologies such as microarray and RNA sequencing, effects of alcohol on transcriptome changes and molecular mechanisms of transcriptomes have been identified in large-scale studies. Recently, gene expression profiling and network analysis after acute and chronic ethanol exposure have been performed in vitro and in vivo [6,7,8,9]. Previously, we have reported that treatment with 50 mM ethanol during proliferation and differentiation of neural stem cells induces upregulation of small heat shock protein genes and alterations of Wnt signaling molecules related to brain development [7,8]. Studies using animal models also revealed that chronic ethanol abuse induces changes of myelin gene expression, chromatin remodeling, and altered synaptic transmission in several regions of the brain, including the hippocampus and prefrontal cortex [10,11].

The human genome is primarily composed of non-coding RNAs (ncRNAs) that are not translated into proteins. These ncRNAs are classified as small RNAs (17–25 nucleotides) including microRNAs (miRNAs), mid-size RNAs (20–200 nucleotides), and long ncRNAs (lncRNAs, more than 200 nucleotides) according to nucleotide number [12]. Among the three types of ncRNAs, miRNAs are reported to mediate post-transcriptional gene silencing [13], although lncRNAs were previously considered transcriptional noise [14]. Recently, as lncRNAs have been revealed to regulate many biological processes including epigenetic and transcriptional modification [15,16], they are emerging as potential therapeutic targets in many diseases. In particular, since it has been reported that dysregulation of lncRNAs is associated with development of many cancers including glioblastoma, hepatocellular carcinoma, and liver and lung cancers [17], molecular mechanisms of lncRNAs are being studied in cancers.

ncRNAs are expressed in the brain and are associated with neuronal diseases caused by excess alcohol use [18]. In particular, alterations of endogenous miRNAs associated with chronic ethanol use have been well-studied [19,20,21,22]. On the other hand, until recently, lncRNAs were mainly examined in cancer-related studies focusing on cancer development mechanisms. A small number of lncRNAs has been associated with alcohol-related diseases, and studies of their molecular mechanisms are in very early stages [23,24,25].

When the effects of ethanol are investigated using cell lines, it is important to select ethanol concentrations similar to those affecting animal models or humans. Zorumski et al. [26] have described that low ethanol concentrations (5–10 mM) are similar to the effect of one or two standard drinks (each with ~12 g of alcohol) on glutamate receptors. Additionally, high ethanol concentrations (60 mM and above) can partially inhibit N-methyl-D-aspartate receptors (NMDARs), which mediate fast neurotransmission. A mouse neuroblastoma cell line Neuro2a is widely used to investigate axonal growth, neuronal differentiation, and signaling pathways because Neuro2a cells have the ability to differentiate into neurons [27,28,29]. As Neuro2a cells are also able to metabolize alcohol and alcohol metabolism is associated with damage to neural tissue due to alcohol, these cells are a good in vitro model for studying the neurotoxicity of alcohol [30,31,32,33]. In addition, previous studies identified that Neuro2a cells exposed to 100 mM ethanol for three days did not show any significant cell death [31] and regarded 100 mM ethanol concentration in Neuro2a cells as a pharmacologically relevant concentration of 400 mg/dL [30].

Based on these circumstances, we hypothesized that changes in the expression of lncRNAs and mRNAs would be observed in neuronal cells exposed to ethanol. Therefore, after acute and chronic ethanol (100 mM) treatment of Neuro2a cells, we profiled coding and noncoding RNA expression using total RNA-Seq and functional annotation of differentially expressed genes (DEGs) and analyzed their protein networks. We found that expression of *Cebpd* was altered by ethanol and affected the expression of *Rnu3a*.

## 2. Results

### 2.1. Identification of Genes Differentially Expressed in Response to Ethanol

To assess the effects of ethanol on proliferation of Neuro2a cells, we treated cells with 50 mM or 100 mM ethanol for 24 and 72 h. Neither concentration of ethanol affected cell proliferation at 24 h, but 100 mM ethanol induced significant reduction of cell proliferation at 72 h without cellular damage under the microscope (Figure 1a,b). Based on this result, we analyzed expression changes of mRNA and lncRNAs in Neuro2a cells exposed to 100 mM ethanol for 24 and 72 h using total RNA-Seq. After mining significant data from the control, 100 mM ethanol-treated cells for 24 h (E24h), and 100 mM ethanol-treated cells for 72 h (E72h) using log2 transformation (fold change cutoff of 3), a heat map of 1773 total RNAs was constructed using hierarchical clustering analysis (Figure 1c). The gene expression patterns were similar between control and E24h groups, both of which were opposite to that in the E72h group. We also created Venn diagrams to visualize specific or overlapped transcripts among comparison pairs (E24h vs. Control, E72h vs. Control, and E24h vs. E72h) (Figure 1d). Of the 1773 transcripts, 70 and 1339 were upregulated or downregulated more than three-fold in the E24h and E72h groups, respectively, compared to the control group. When comparing E24h and E72h groups, 1512 transcripts were differentially expressed most strongly among comparison pairs. mRNAs selected from total RNA-Seq analysis based on gene function annotation were validated using RT-qPCR and are presented in Figure 1e.

### 2.2. Functional Annotation and Pathway Network of DEGs

We performed gene ontology (GO) enrichment and functional annotation analysis using g:Profiler and identified three GO categories (biological process [BP], cellular component [CP], and molecular function [MF]). As a result of enrichment on the top 20 GO terms that satisfied the adjusted *p* values in the BP category, genes related to metabolic and biosynthetic processes were differentially expressed in the E72h group compared to the control and E24h groups, respectively (Figure 2a,b, and Appendix A). On the other hand, after the most significant GO term was selected per comparison pair, the enriched GO terms in the BP category obtained among comparison pairs are presented in Figure 2c and were compared simultaneously (Appendix A). Nucleic acid metabolic process and nucleobase-containing compound metabolic process were the respective first and second most significant GO terms in all comparison pairs. In addition, as a result of selecting the most significant GO term in the BP category per comparison pair according to term size, the GO term DNA conformation change was most significant and was the largest DEG in all comparison pairs (Figure 2d and Appendix A).

To construct protein–protein interactions of DEGs among the three groups, we used the STRING database. When comparing DEGs between the E24h and control groups, up-regulated genes at E24h were responsible for inflammatory responses and signaling (Figure 3a), while down-regulated genes were involved in response to heat and protein folding (Figure 3b). In comparisons of E72h and control groups, up-regulated genes at E72h were involved in cell fate commitment and ventral axis formation (Figure 3c), while down-regulated genes were related to covalent chromatin modification, the toll-like receptor signaling pathway, and cell cycle/cell division (Figure 3d). When comparing DEGs between E24h and E72h groups, genes related to alcoholism and chromatin silencing were upregulated in E72h, while genes involved in pexophagy, DNA repair, and microtubule bundle formation were downregulated in E72h (Appendix A).

### 2.3. Identification of lncRNAs Differentially Expressed in Response to Ethanol

Among 1773 total RNAs (coding and noncoding RNAs), 514 lncRNAs that satisfied the fold change cutoff of 3 were sorted, and a heat map was constructed for them using hierarchical clustering analysis (Figure 4a). The expression patterns of lncRNAs among groups were similar to the expression patterns of total RNAs. On the other hand, 127 and 413 lncRNAs were upregulated or downregulated in E24h and E72h groups, respectively, compared to the control group, as visualized in Venn diagrams showing a number of specific or overlapped lncRNAs (Figure 4b). Finally, 78 lncRNAs were upregulated or downregulated in both E24h and E72h groups compared to the control group. Figure 4c presents lncRNA lists selected from total RNA-Seq analysis to validate using RT-qPCR.

### 2.4. Validation of mRNAs and lncRNAs Based on RT-qPCR

To validate the expression levels of mRNAs and lncRNAs identified by total RNA-Seq analysis, we selected 11 mRNAs and six lncRNAs and assayed them using RT-qPCR. Among genes related to nucleic acid metabolic process (*Atm*, *Bmpr2*, *Rb1*, *Hes7*, and *Trib3*), *Atm*, *Bmpr2*, and *Rb1* were significantly downregulated in the E72h group compared to E24h and control groups, while *Trib3* was significantly upregulated in the E72h group compared to the E24h and control groups (Figure 5a). Among genes involved in cellular macromolecule metabolic process (*Bmpr1a*, *Cebpd*, and *Spdef*) and regulation of cellular metabolic process (*Braf*, *Cnr1*, and *Pik3c2a*), *Bmpr1a*, *Cebpd*, *Braf*, *Cnr1*, and *Pik3c2a* were significantly downregulated in the E72h group compared to the E24h and control groups, while *Spdef* was significantly upregulated in both the E24h and E72h groups compared to the control group. After validating the expression levels of 6 lncRNAs (*Tbrg3*, *Kcnq1ot1*, *Tug1*, *Xist*, *Rnu3a*, and *4930507D05Rik*), we found that *Tbrg3*, *Kcnq1ot1*, *Tug1*, and *Xist* were significantly downregulated in the E72h group compared to the control and E24h groups (Figure 5b). However, *Rnu3a* and *4930507D05Rik* were significantly upregulated in the E72h group compared to the control and E24h groups. Therefore, the expression patterns of all validated mRNAs and lncRNAs were similar to those observed in total RNA-Seq.

### 2.5. Knock-Down of Cebpd

Cebpd, a transcription factor, regulates pro-apoptotic gene expression [34]. To determine whether Cebpd affects the expression of *Rnu3a*, we predicted putative promoter sequences of the *Rnu3a* gene using the Pscan and JASPAR databases. There was a Cebpd binding motif in the promoter region of *Rnu3a* (Figure 5c). Based on this result, we induced knock-down of Cebpd in Neuro2a cells using siRNA targeting *Cebpd* (si-Cebpd). Knock-down of *Cebpd* led to a significant decrease in the expression of *Rnu3a* and *Cebpd* (Figure 5d,e). Therefore, Cebpd can act as a transcription factor to regulate the expression of lncRNA *Rnu3a*.

## 3. Discussion

Chronic alcohol abuse causes neurobiological and neuropathological alterations including changes of neurotransmitter systems in the brain, resulting in deficits of cognitive and learning memories and neurodegenerative diseases. These alterations highlight the need to reveal the diverse mechanisms and pathways that are involved in alcohol-related diseases. In addition, the development of large-scale profiling technology allowed identification of genes and mechanisms related to such diseases. Using this technology, we discovered a large number of coding and noncoding RNAs that are differentially expressed according to ethanol exposure and analyzed their associated gene functions. 

In the present study, proliferation of Neuro2a cells exposed to 100 mM ethanol for 72 h was significantly decreased, showing that the dosage of ethanol negatively affects cell proliferation during chronic treatment. In accordance with our results, a previous study observed that the proliferation rate of Neuro2a cells treated with 100 mM ethanol for 72 h was significantly lower than that of cells treated with 100 mM ethanol for 0, 24, and 48 h [24]. Cell viability in cells treated with a set ethanol concentration did not show significant differences at 0, 24, 48, and 72 h but showed a significant decrease at 96 h. Other studies verified that 100 mM ethanol mimics alcoholic injury in hepatocytes and cardiomyocytes without significant effects on cell viability at 24 and 48 h [35,36]. Taken together, these findings suggest that 100 mM ethanol is appropriate for investigating effects of alcohol on alterations of gene expression in brain-derived cell lines in vitro. 

In our study, total RNA expression patterns in the E24h group were similar to those in the control group, while the E72h group showed opposite expression patterns. These results indicate that chronic treatment with 100 mM ethanol induces dramatic changes of gene expression patterns, including noncoding RNAs. Chronic exposure to ethanol causes alterations of metabolic systems in the body. Alcohol dehydrogenase (ADH) and aldehyde dehydrogenase (ALDH) metabolize ingested alcohol [37] and promote the breakdown of alcohol into acetaldehyde (a highly toxic carcinogenic) and acetate. These by-products affect the metabolic processes of cells in tissues. In particular, the metabolites of alcohol can exert psychoactive effects in the body for longer durations than can alcohol itself [38,39]. In GO analysis in the present study, the E72h group showed significant changes in genes related to various metabolic processes, including nucleic acid metabolic process, cellular macromolecule metabolic process, and regulation of cellular metabolic process, compared to the E24h and control groups. Taken together, these results suggest that alcohol affects metabolic processes both in vitro and in vivo. 

In the present study, among nucleic acid metabolic process-related genes (*Atm*, *Bmpr2*, *Rb1*, *Hes7*, *Trib3*) whose expression is changed by ethanol, *Atm* in particular was significantly down-regulated by chronic treatment with ethanol. ATM, a canonical DNA damage checkpoint protein, is activated in mitosis due to DNA damage and helps maintain genome stability [40]. ATM has also been reported to prevent oxidative stress by stimulating pexophagy, a type of selective autophagy, in response to hydrogen peroxide [41]. Based on previous studies and our results, we hypothesize that chronic exposure to ethanol, an oxidative stressor, can induce genome instability through downregulation of *Atm* expression in brain cells. On the other hand, Trib3 is known to be upregulated by a variety of stresses [42]. A previous study reported that Trib3 was increased in rat cortical neurons during endoplasmic reticulum (ER) stress caused by kainic acid, and that knock-down of *Trib3* by siRNA decreased apoptosis of neurons due to ER stress [43]. Similar to the results of previous studies, in our study, *Trib3* mRNA was significantly up-regulated in Neuro2a cells by chronic ethanol treatment. Taken together, our results suggest that chronic exposure to ethanol can increase ER stress and autophagy in cells due to upregulation of *Trib3* mRNA, resulting in negative effects on cell proliferation. However, further research is needed to determine whether ethanol directly affects cell proliferation through Trib3-related mechanisms. 

In our research, mRNAs of Cebpd and Spdef, proteins responsible for cellular macromolecule metabolic process, increased after 72 h of ethanol treatment. In particular, ethanol-treated *Spdef* was upregulated at both 24 h and 72 h. Cebpd, classified as a leucine zipper transcription factor, is a stress response gene that is triggered by various stimuli [44]. Its function varies depending on cell type or situation. A previous study reported that expression of Cebpd in *Cebpd*-deficient chronic myelogenous leukemia cell lines led to increased apoptosis as well as G0/G1 proliferative arrest [34]. On the other hand, other studies found that Cebpd is highly expressed in astrocytes and promotes up-regulation of SOD1 in astrocytes, resulting in astrocyte resistance against reactive oxygen species (ROS) [45]. Astrocytes activated by ROS induce neurological damage [46]. In addition, up-regulation of Cebpd has been observed in a mouse model of Alzheimer’s disease (AD), showing a correlation between Cebpd and AD [47]. Our observation that chronic exposure to ethanol led to increased *Cebpd* in Neuro2a cells is consistent with the results of previous studies. Therefore, up-regulation of *Cebpd* due to ethanol might increase the risk of neurological impairments or neurodegenerative brain disease. On the other hand, mRNA of Spdef, an epithelium-specific ETS (ESE) transcription factor, has been reported to be highly expressed in normal human tissues such as stomach, colon, breast, and prostate [48]. In addition, the expression and role of Spdef have been studied mainly in relation to tumor progression in colorectal, hepatocellular, bladder, and prostate cancers. Spdef seems to be a double-edged sword as it promotes or suppresses the progression of cancer depending on the type of cancer [48,49,50]. Therefore, our study is the first to demonstrate that ethanol induces the expression of *Spdef* in brain-derived cells and to highlight the importance of studying the effects of Spdef and alcohol on the brain.

Cannabinoid receptor 1 (CB1) encoded by *CNR1* is abundant in the central nervous system. Deficiencies of the endocannabinoid signaling pathways in which CB1 is presynaptically activated by cannabinoids are more vulnerable in the context of alcohol dependence [51]. A previous study has found that acute alcohol administration in healthy social drinkers results in increases of CB1, but chronic heavy drinking in alcohol-dependent patients induced decreases of CB1 [51]. On the other hand, our observation that chronic ethanol treatment in vitro promoted down-regulation of *Cnr1* is in accordance with previous research using animal models finding that chronic exposure to ethanol induces reduction of *Cnr1* in the hippocampus [52,53]. These results imply that prolonged exposure to alcohol in humans, in vivo and in vitro, gives rise to alterations of endocannabinoid signaling pathways by inhibiting expression of *CNR1* mRNA. 

lncRNAs are involved in various gene regulation and biological processes such as translation, nuclear and cytoplasmic trafficking, cell proliferation, differentiation, and apoptosis [54,55]. Regulation of their expression and functions by diverse factors affects cell fate. In our study, exposure of Neuro2a cells to ethanol promoted changes of many lncRNAs including *Kcnq1ot1*, *Tug1*, *Xist*, and *Rnu3a*. Among the altered lncRNAs, *Kcnq1ot1* was significantly increased at 24-h after ethanol treatment but decreased at 72-h after ethanol treatment. In previous studies, down-regulation of *Kcnq1ot1* inhibited β-cell proliferation, resulting in impaired insulin synthesis [56], and its overexpression ameliorated osteogenic differentiation in human bone mesenchymal stem cells infected by bacteria [57]. Considering previous research, *Kcnq1ot1* plays a role in helping cells proliferate and differentiate. This is line with our observation that ethanol caused reduced cell proliferation and *Kcnq1ot1* expression after 72-h ethanol treatment. However, as 24-h ethanol treatment promoted up-regulation of *Kcnq1ot1* and did not affect cell proliferation, ethanol differentially mediates *Kcnq1ot1* expression depending on time. 

In this research, the expression of lncRNA *Tug1* in Neuro2a cells increased after 24-h ethanol treatment but decreased after 72-h ethanol treatment. On the other hand, other teams have reported that knock-down of *Tug1* induced decreases of cell apoptosis in oxygen and glucose deprivation (OGD)-induced HT22 cells (mouse hippocampal neuronal cell line) [58] and resolved suppression of proliferation and stimulation of inflammation and apoptosis in OGD/reoxygen (OGD/R)-induced Neuro2a cells [59]. These findings are similar to our results showing increases of *Tug1* in Neuro2a cells exposed to ethanol for 24 h. However, in our study, 72-h ethanol treatment promoted highly decreased *Tug1* expression. Given that OGD-induced HT22 cells experienced deprivation for 1 h and were then cultured in normal medium containing FBS for 24 h [58], and OGD/R-induced Neuro2a cells also experienced the OGD environment for 24 h, followed by immediate reoxygenation for 24 h [59]. This recovery environment of cells re-exposed to normal culture condition after deprivation might have affected the *Tug1* expression differently from our study in which Neuro2a cells were exposed to ethanol in normal culture conditions. As *Tug1* expression patterns under ethanol-treated conditions were observed in reverse depending on time, further research is needed to elucidate how changes of *Tug1* expression and function under ethanol-treated condition affect cells. 

In this research, lncRNA *Xist* was significantly increased after 24-h ethanol treatment but highly decreased after 72-h ethanol treatment. *Xist* mediates X chromosome inactivation (XCI) and also plays an important role in cell growth and development [60]. *Xist* as a lncRNA regulating coding genes except for XCI has been widely studied in cancer. According to review by Wang et al. [60], *Xist* plays an oncogenic role in bladder, breast, and colorectal cancers, hepatocellular carcinoma, and glioblastoma. On the contrary, *Xist* inhibits cell proliferation in triple-negative breast cancer, a subtype of breast cancer, and deficiency of salivary *Xist* expression promotes risk of oral squamous cell carcinoma [61]. *Xist* has been demonstrated to alleviate Parkinson’s disease by protecting dopaminergic neurons in animal models [62]. Based on previous studies, *Xist* is thought to show tissue specificity as it acts not only as an oncogenic repressor in most cancers, but also as a suppressor in some cancers. In addition, since the expression pattern of *Xist* was different depending on ethanol treatment time in our study, the change in *Xist* expression might be related to cell state according to ethanol exposure time.

However, we identified that the expression of *Rnu3a*, a member of the small nuclear RNA (snRNA) family, is highly increased in Neuro2a by chronic ethanol treatment. snRNAs are involved in ribosome biosynthesis, nuclear maturation of primary transcripts, and gene expression regulation [63]. *Rnu3a*, also belonging to the ncRNAs, is expected to play these roles, although its function has not been clearly elucidated. Previous research has showed that *Rnu3a* was reduced in the nucleus pulposus (NP) of aged mice compared to that of young mice, suggesting that *Rnu3a* has potential as a biomarker of NP degeneration [64]. However, considering that the previous result was obtained by microarrays using only two to three–3 mice in each group, it is necessary to validate these results by performing qPCR or repeating the animal study in a larger sample to address individual variation, unlike in vitro experiments. On the other hand, based on our observation that chronic ethanol treatment in Neuro2a increased expression of *Rnu3a,* which has the binding site (TTGCACAA) of transcription factor Cebpd, we further verified that Cebpd was involved in *Rnu3a* expression. Knock-down of *Cebpd* resulted in inhibition of the expression of both *Cebpd* and *Rnu3a*. Therefore, Cebpd as a transcription factor is thought to control transcription of *Rnu3a* by binding to the binding motif of *Rnu3a* DNA. In addition, under ethanol exposure, Cebpd is thought to affect cells through regulation of *Rnu3a*. In the future, studies of how Cebpd affects cells exposed to ethanol by controlling *Rnu3a* are necessary.

In conclusion, we confirmed that chronic exposure of Neuro2a cells to 100 mM ethanol induces decreased cell proliferation. We found that ethanol treatment led to changes in the expression of many mRNAs and lncRNAs in Neuro2a cells. Among the DEGs, those associated with metabolic processes including nucleic acid metabolic process were mainly changed by ethanol. *Atm* and *Cnr1* decreased but *Trib3*, *Cebpd*, and *Spdef* increased after ethanol treatment for 72 h. On the other hand, lncRNAs *Kcnq1ot1*, *Tug1*, and *Xist* were changed by ethanol, and *Rnu3a* was greatly increased by chronic ethanol treatment. Based on these observations, we demonstrated that Cebpd, a transcription factor, regulates the expression of *Rnu3a*. To the best of our knowledge, our research is the first to perform expression profiling of both mRNAs and ncRNAs in brain-derived cells after acute and chronic ethanol treatment. Since we investigated expression changes of mRNAs and ncRNAs due to ethanol in a neuronal cell line in vitro, we should be careful in generalizing our results to the human brain, which consists of non-neuronal cells as well as neuronal cells. Nevertheless, our results pave the path to a better understanding of cellular and molecular mechanisms related to coding and noncoding RNAs in in vitro models of acute and chronic exposure to alcohol.

## 4. Materials and Methods

### 4.1. Cell Culture

Neuro2a cells purchased from the American Type Culture Collection were cultured in Dulbecco’s modified Eagle’s medium (DMEM; Thermo Fisher Scientific, Waltham, MA, USA) with 10% fetal bovine serum (FBS; Thermo Fisher scientific, Waltham, MA, USA) and 100 units/mL penicillin plus 100 mg/mL streptomycin at 37 °C with 5% CO_2_. After the cells reached 70–80% confluence, they were subcultured using 0.25% trypsin/EDTA (Thermo Fisher Scientific, Waltham, MA, USA).

### 4.2. Alcohol Treatment and Cell Proliferation Assay

Cell proliferation was examined using the Cell Counting Kit-8 (CCK-8; Dojindo, Kumamoto, Japan) according to the manufacturer’s instructions. Prior to alcohol treatment, Neuro2a cells were seeded in 96-well plates at a density of 5000 cells per well and incubated at 37 °C and 5% CO_2_ for 24 h. To evaluate the effects of alcohol on Neuro2a cell proliferation, 100 μL of fresh medium with or without alcohol (50 and 100 mM) was applied to each well and incubated at 37 °C with 5% CO_2_ for 24 and 72 h. For 72 h alcohol treatment, alcohol and fresh medium were applied every day. At 24 and 72 h after alcohol treatment, 10 μL CCK-8 solution was added to each well containing 100 μL culture medium, and the plate was incubated for 2 h at 37 °C in 5% CO_2_ incubator. Cell proliferation was quantified by measuring the absorbance at 450 nm using a microplate reader (Power Wave XS2, BioTeck, Winooski, VT, USA).

### 4.3. Total RNA-Seq Library Preparation and Sequencing

Neuro2a cells treated with 100 mM alcohol were harvested at 24 and 72 h. Total RNA was extracted from 3 groups of Neuro2a cells using Trizol reagent: Control (2 samples), E24h (2 samples), and E72h (2 samples). The integrity of the total RNA was assessed using a 2100 Bioanalyzer (Agilent Technologies, Palo Alto, CA, USA), and the RNA integrity number (RIN) of the RNA was 10. RNA libraries were constructed using the TruSeq Stranded Total RNA with Ribo-Zero H/M/R_Gold kit (Illumina, San Diego, CA, USA) according to the manufacturer’s instructions. The constructed libraries were 101-bp paired-end sequenced using a NovaSeq 6000 system (Illumina, San Diego, CA, USA), and sequence quality was verified with FastQC (version 0.11.7). Before analysis, Trimmomatic (version 0.38) [65] was used to remove adapter sequences and trim bases with quality lower than 3 from both ends of the reads. Bases that did not qualify for window size = 4 and mean quality = 15 were filtered by the sliding window trim method, and reads less than 36-bp in length were excluded to produce high-quality data. 

### 4.4. Differential Gene Expression Analysis and Enrichment

The preprocessed reads were aligned to the mouse genome (*Mus musculus*, mm10) downloaded from the NCBI using HISAT v2.1.0 [66]. The relative abundance of genes was calculated as read counts or fragments per kilobase million (FPKM) mapped value per sample using StringTie v2.1.3b [67]. We performed statistical analysis to identify differentially expressed genes using estimates of abundances for each gene in the control and alcohol-treated samples. Genes with more than one zeroed Read Count value in the samples were excluded. Filtered data were log2-transformed and subjected to RLE normalization. The statistical significance of differential expression data was determined using nbinomWaldTest in DESeq2 [68] and fold change, in which the null hypothesis was no difference among groups. The false discovery rate (FDR) was controlled by adjusting *p* values using the Benjamini–Hochberg algorithm. Genes and ncRNAs that satisfied |fold change| ≥ 3.0 in at least one among comparison pairs were selected. For differentially expressed total RNA and lncRNA sets, hierarchical clustering analysis was performed using complete linkage and Euclidean distance as a measure of similarity. Gene-enrichment and functional annotation analyses of DEGs were performed based on gProfiler (https://biit.cs.ut.ee/gprofiler/orth, accessed on 10 August 2021), and their pathways and networks were analyzed using the STRING database (https:string-db.org, accessed on 21 August 2021).

### 4.5. RT-Quantitative PCR (RT-qPCR)

We performed RT-qPCR to validate genes and lncRNAs differentially expressed in total RNA-Seq. Neuro2a cells treated with 100 mM alcohol were harvested at 24 and 72 h. Total RNA was extracted from the cells using TRIzol reagent (Thermo Fisher Scientific, San Jose, CA, USA) and reverse transcribed into cDNA using the Superscript IV First-Strand Synthesis System (Thermo Fisher Scientific, San Jose, CA, USA) according to the manufacturer’s instructions. Details of the qPCR method have been described previously [7]. Three independent qPCR experiments were performed to guarantee reliable results for all samples from each group (Control, E24h, and E72h groups). The expression of the DEGs and DEncRNAs in each sample was normalized to that of *GAPDH*. The relative expression differences among the three groups were calculated using the 2^−ΔΔCT^ method. The primers used for amplification of candidate genes are presented in Appendix A. 

### 4.6. Knock-Down of Cebpd

To identify novel candidate sites of lncRNAs for Cebpd, we searched for Cebpd binding motifs on lncRNA genes using the Pscan database. After selecting *Rnu3a* with a corrected *p*-value less than 0.05, we matched Cebpd binding motif and *Rnu3a* sequences using the JASPAR database. Small interference RNAs (siRNAs) for *Cebpd* and negative controls were generated using Genolution (Seoul, Korea). The siRNA sequences for *Cebpd* (si-Cebpd) are 5’-CGACUUCAGCGCCUACAUUUU-3’ and 5’-AAUGUAGGCGCUGAAGUCGUU-3’. The siRNA sequences for the negative control (si-NC) are 5′-CCUCGUGCCGUUCCAUCAGGUAGUU-3′ and 5′-CUACCUGAUGGAACGGCACGAGGUU-3′. Neuro2a cells were seeded on a 6-well plate and incubated at 37 °C and 5% CO_2_ to 60–70% confluence. After that, 100 pmol of siRNAs was transfected into the cells using Lipofectamine 3000 transfection reagent (Thermo Fisher Scientific, Waltham, MA, USA) according to the manufacturer’s instructions. The cells were incubated at 37 °C and 5% CO_2_. After 48 h, the cells were harvested, and total RNA was extracted using TRIzol reagent (Thermo Fisher Scientific, San Jose, CA, USA). After being reverse transcribed into cDNA, qPCR for analyzing expression changes of *Cebpd* and *Rnu3a* was performed in the same manner as described above (RT-qPCR).

### 4.7. Statistical Analysis

Statistical analyses were conducted with GraphPad Prism 8 software (San Diego, CA, USA). All data obtained from cell proliferation assays and RT-qPCR were expressed as mean ± standard error of the mean (SEM). Differences among control, E24h, and E72h groups for data obtained from cell proliferation assay and RT-qPCR were analyzed with one-way ANOVA followed by the HSD test. *p* < 0.05 was considered statistically significant.

## Figures and Tables

**Figure 1 ijms-23-07294-f001:**
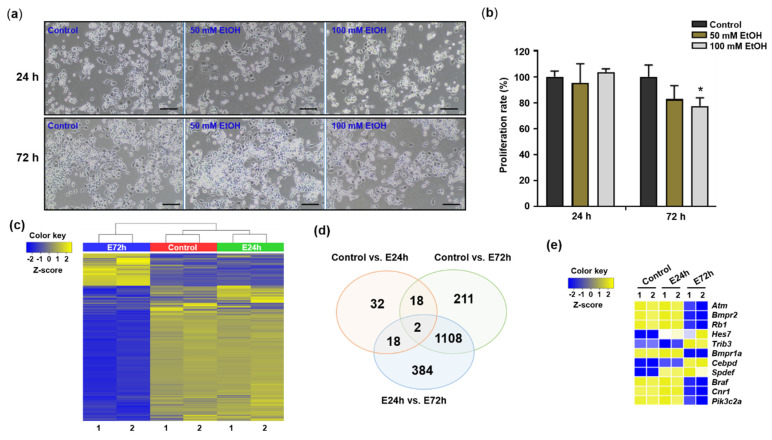
Changes of cell proliferation and gene expression in Neuro2a cells in response to ethanol. (**a**) Morphology of Neuro2a cells exposed to ethanol at 100× magnification. The cells were exposed to ethanol (50 and 100 mM) for 24 (upper panels) and 72 h (lower panels). Scale bar = 100 µm. (**b**) Proliferation of Neuro2a cells exposed to ethanol (EtOH). Proliferation rates of the cells exposed to ethanol (50 and 100 mM) for 24 and 72 h were evaluated. Ethanol treatments were performed as three independent replicates to guarantee reliable results. To compare the differences among control and ethanol-treated cells, one-way ANOVA and Tukey’s post-hoc tests were used. *Significantly different from control (* *p* < 0.05). (**c**) Heatmap generated from hierarchical clustering analysis of differentially expressed transcripts among control and ethanol-treated (E24h and E72h) groups. (**d**) Venn diagram showing the overlap of transcripts among control, E24h, and E72h groups. (**e**) mRNAs selected from the total transcripts for RT-qPCR. Cells exposed to 100 mM ethanol for 24 h, E24h; cells exposed to 100 mM ethanol for 72 h, E72h.

**Figure 2 ijms-23-07294-f002:**
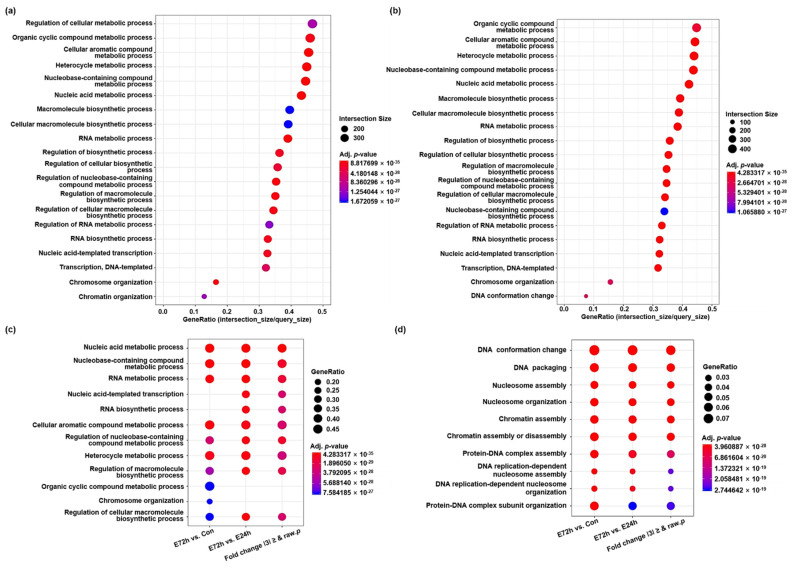
Gene ontology (GO) annotation of DEGs among control and ethanol-treated groups. Enrichment and functional annotation of DEGs among control, E24h, and E72h groups were performed using g:Profiler. (**a**) Top 20 enriched GO terms in the biological process (BP) category of genes expressed differentially in E72h compared to the control. (**b**) Top 20 enriched GO terms in the BP category of genes expressed differentially in E72h compared to E24h. (**c**) Top 10 GO terms in the BP category obtained among comparison pairs. (**d**) Top 10 GO terms in the BP category obtained per comparison pair according to term size. Query size: the number of unique DEGs annotated to the data source. Intersection size: the number of unique DEGs annotated to the term ID.

**Figure 3 ijms-23-07294-f003:**
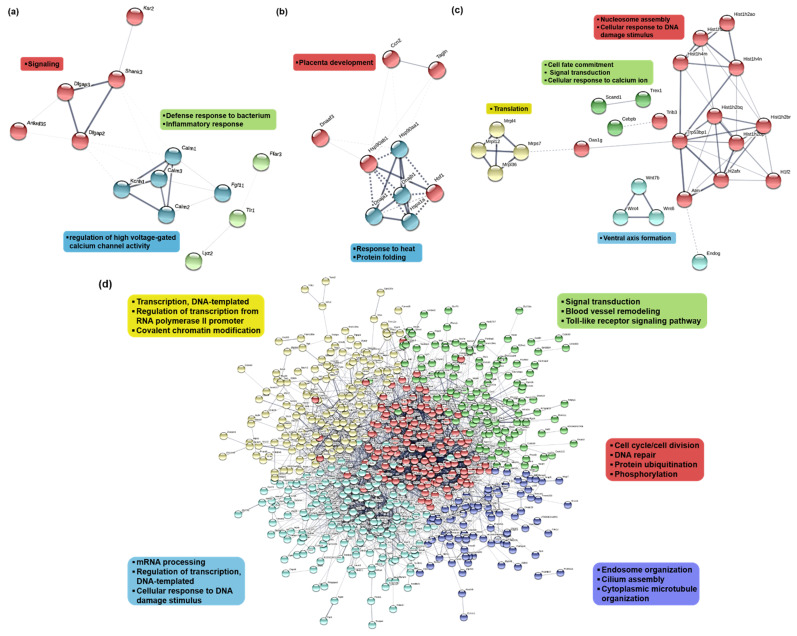
Protein interaction network of DEGs in Neuro2a cells in response to ethanol. Protein interaction networks of DEGs were mapped using the STRING database. (**a**,**b**) Protein interaction network of genes upregulated and downregulated, respectively, in E24h compared to the control. (**c**,**d**) Protein interaction network of genes upregulated and downregulated, respectively, in E72h compared to the control. Each protein (node) is connected by a line, the thickness of which depicts confidence in the association (line thickness indicates the strength of the data support).

**Figure 4 ijms-23-07294-f004:**
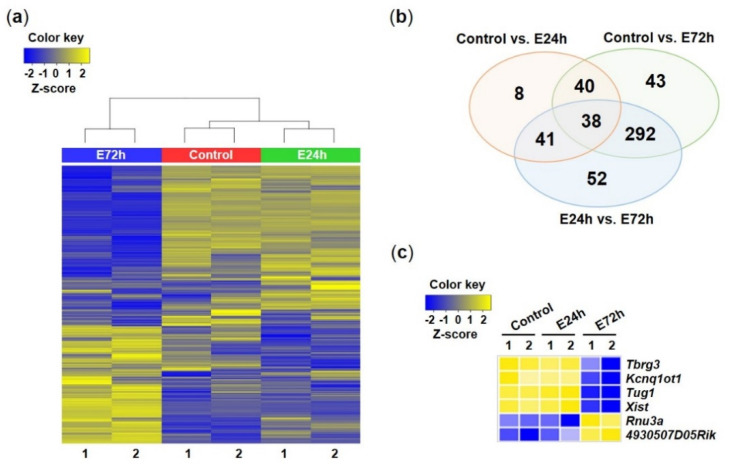
Changes of lncRNAs among control and ethanol-treated groups. (**a**) Heatmap generated from hierarchical clustering analysis of differentially expressed lncRNAs among control and ethanol-treated (E24h and E72h) groups. (**b**) Venn diagram showing overlap of lncRNAs among control, E24h, and E72h groups. (**c**) lncRNAs selected from total transcripts for RT-qPCR.

**Figure 5 ijms-23-07294-f005:**
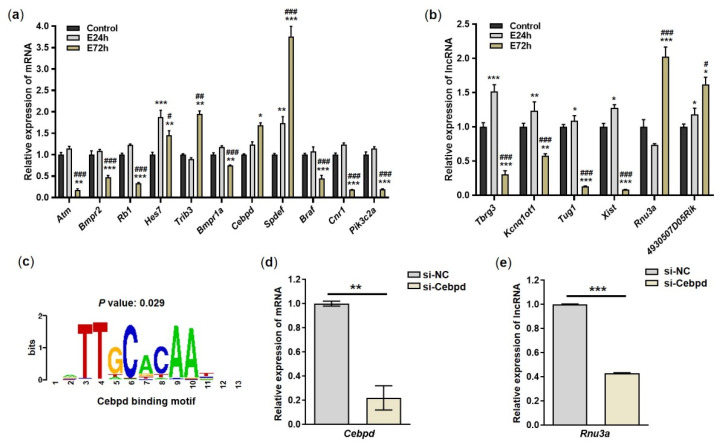
Validation of mRNAs and lncRNAs expressed differentially in Neuro2a exposed to ethanol. After exposing Neuro2a cells to 100 mM ethanol for 24 and 72 h, some mRNAs and lncRNAs were analyzed using RT-qPCR. (**a**) Expression changes of genes related to nucleic acid metabolic processes (*Atm*, *Bmpr2*, *Rb1*, *Hes7*, and *Trib3*), cellular macromolecule metabolic processes (*Bmpr1a*, *Cebpd*, and *Spdef*), and regulation of cellular metabolic processes (*Braf*, *Cnr1*, and *Pik3c2a*). (**b**) Expression changes of lncRNAs. (**c**) Cebpd binding motif in the promoter region of lncRNA *Rnu3a*. (**d**) Expression of *Cebpd* after knock-down using siRNA targeting *Cebpd* (si-*Cebpd*) and in a negative control (si-NC). (**e**) Expression of *Rnu3a* after knock-down of *Cebpd* using si-*Cebpd* and si-NC. *: significantly different from control in (**a**,**b**) and from si-NC in (**d,e**) (* *p* < 0.05, ** *p* < 0.01, and *** *p* < 0.001). ^#^ significantly different from E24h (**a**,**b**) (^#^ *p* < 0.05, ^##^ *p* < 0.01, and ^###^ *p* < 0.001).

## Data Availability

The datasets generated and analyzed in this study are available from the corresponding author upon reasonable request.

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
