# Peer review of "Effects of Ethanol on Expression of Coding and Noncoding RNAs in Murine Neuroblastoma Neuro2a Cells"

_ijms, 2022, doi:10.3390/ijms23137294_

Round 1
Reviewer 1 Report
How were the doses (concentrations) of alcohol selected? It is not clear.
Why did the authors choose Neuro2a cells? It has not been clarified. It is known that these cells, due to passaging since initial collection, can exhibit responses to toxins that differ from those of neuronal cells in a live organism.
The effects of alcohol on the body are complex and not fully understood. Alcohol itself acts as well as its (liver) metabolites. The study of its effect on isolated cells may show its toxic effect, but it does not fully correspond to the physiology of the living organism.
The authors should use an in vivo test to observe the effects on a living organism. Next, isolated tissues/cells could be examined. Moreover, it should be taken into account that the obtained effect will be different after a single administration of alcohol and different after its chronic use.
The authors use the term "chronic administration" for 48, 72, and 96 hours. In experimental studies, chronic administration is defined as administration for several weeks, at least 28 days. Using for many hours can be defined as prolonged alcohol intoxication.
In a well-designed experiment, positive and negative controls should be considered as a reference (standard agonist and antagonist). In the case of alcohol it is difficult, therefore the analysis of data from such an isolated experience, although informative, does not necessarily correspond to real phenomena in a living organism. Our observations show that many genes, including those from the group of "housekeeping genes", are susceptible to the effects of alcohol, which indicates that its action is multidimensional.
The authors used the GAPDH gene for validation. Did this gene meet the conditions for the reference gene? Was is stable under the influence of alcohol? A reference gene is a gene whose expression is stable in all studied conditions. It would ideally have exactly the same expression level in all samples, with only measurement noise added. In practice, it is expected to have a very low variability of expression in each of the tested conditions and an equal average level of expression between all tested conditions. Normalization using a single reference gene is risky as stated in the MIQE (Minimum Information for Publication of Quantitative Real-Time PCR Experiments) guidelines. In addition, appropriate reference genes should not belong to the same pathway or the same gene family.
Reviewer 2 Report
The manuscript describes an RNA sequencing analysis of a neuroblastoma cell line subjected to alcohol (100mM) treatment. As is standard in such analyses, differential expression, gene ontologies and network diagrams linking regulated genes are presented. Whilst I cannot fault in any significant way the presentation of the study, and I can forsee it may be of interest to a specialist as an additional dataset, in my opinion there is very little insight or inventiveness as this really is just a fairly standard sequencing and analysis of a simple treatment on a cell line. I present several very small modifications in my opinion the authors should easily make to the manuscript to tidy some very minor issues. I also suggest several analyses not present that could be considered should the editor feel it appropriate push beyond the standard analysis presented here:
1) Cell morphology is described but not presented in Figure 1. I would include this.
2) In materials and methods (4.2), please clarify whether old media was removed when fresh media + alcohol was added (I assume as much) as this will affect the dosage.
3) The reader will immediately tend to link these results to the systemic effect of alcohol, yet applying 50 or 100mM alcohol to the media of Neuro2a cells is hardly an approximation of alcohol use. I would recommend the authors pay particular attention to this aspect in the intro / discussion with a specific evaluation of what such a study can (and cannot) tell us.
Suggestions that may increase the significance of this report (without requiring extensive experimentation):
4) Gene expression networks are driven by key proteins (especially transcription factors), splicing factors and ncRNAs like miRNAs. By their nature of affecting many downstream genes, one would anticipate large roles played by key members of such gene families to drive the genetic profile. Further analysis would be encouraged to identify these and predict their downstream effects
5) Related to this, appying a tool such as EISA (Gaidatzis, Nat Biotech 2015) could be beneficial to further tease out transcriptional vs post-transcriptonal effects. I return to this as I really don't see what insightful or useful information is really obtainable from simply providing gene ontologies and highly populated edge/node networks.
6) I admit i'm not sure how one could address this, but is what is reported really an effect of alcohol (as it relates to the effect it has consuming it) or is it really just a series of stress effects any cell would do when provided the high concentration of a toxin / solvent? I can imagine a similar paper constructed by throwing DMSO onto cells for example and monitoring gene expression.
Round 2
Reviewer 2 Report
I have left comments to the editor to the effect that the study is perfectly well presented but severely limited in its ambition and level of interest. No opportunity was taken to try to expand this (without relying on additional experiments), such as re-evaluating networks with a perspective of the possible role of key regulators such as transcription actors or splicing factors. No additional discussion of value regarding the link between this work and any real world application is made.